# GEOMETRYZERO: ADVANCING LLM FOR GEOMETRY SOLVING VIA GROUP CONTRASTIVE POLICY OPTIMIZATION

## ABSTRACT

Recent advances in large language models (LLMs) have demonstrated remarkable capabilities in mathematical reasoning, amid which geometry problem solving remains a challenging area where auxiliary construction plays a enssential role. Existing approaches either achieve suboptimal performance or rely on colossal LLMs (e.g., GPT-4o), incurring massive computational costs. We posit that reinforcement learning with verifiable reward (e.g., GRPO) offers a promising direction for training smaller models that effectively combine auxiliary construction with robust geometric reasoning. However, directly applying GRPO to geometric reasoning presents fundamental limitations due to its dependence on unconditional rewards, which leads to indiscriminate and counterproductive auxiliary constructions. To address these challenges, we propose **Group Contrastive Policy Optimization** (**GCPO**), a novel reinforcement learning framework featuring two key innovations: (1) *Group Contrastive Masking*, which adaptively provides positive or negative reward signals for auxiliary construction based on contextual utility, and a (2) *Length Reward* that promotes longer reasoning chains. Building on GCPO, we develop GeometryZero, a family of affordable-size geometric reasoning models that judiciously determine when to employ auxiliary construction. Our extensive empirical evaluation across popular geometric benchmarks (w.r.t. Geometry3K and MathVista) demonstrates that GeometryZero models consistently outperform RL baselines (e.g. GRPO, ToRL) across various benchmarks.

## 1 INTRODUCTION

Recent advancements in large language models (LLMs) have demonstrated remarkable performance across domains (Ouyang et al., 2022; Team, 2024; DeepSeek-AI et al., 2025) including mathematics (Shao et al., 2024). Among them geometry problem solving is deemed as a challenging task, which requires both perception of visual contexts (i.e., geometric diagrams) and complex reasoning (Lu et al., 2021; Kazemi et al., 2023). Existing training methods either utilize massive annotated data for supervised learning (Gao et al., 2023) or focus on algebraic-level formal deviation (Brehmer et al., 2023). This makes current models show unsatisfying performance on this domain and lack self-correction capabilities in their reasoning chains due to their reliance on annotation quality (Lu et al., 2024).

Another sequence of works focuses on auxiliary lines, which are valuable either when diagrams are inherently complex or when the problem's intrinsic properties benefit from such constructions, significantly reducing problem solving difficulty (Chervonyi et al., 2025). Several works including Hu et al. (2024); Wang et al. (2025) have attempted to enhance visual language models' utilization of contexts through modifying formal languages (e.g., code) for auxiliary construction, thereby improving their reasoning capabilities on complex geometry problems. Existing works validate that transforming visual contexts into formal languages and leveraging LLM yields better reasoning (Yang et al., 2025). AlphaGeometry2 (Chervonyi et al., 2025) also employs LLMs for auxiliary construction. However, these approaches rely on prompting or training colossal models (e.g., Gemini, GPT-4o), which incur expensive computational costs that limit their real-world deployment.

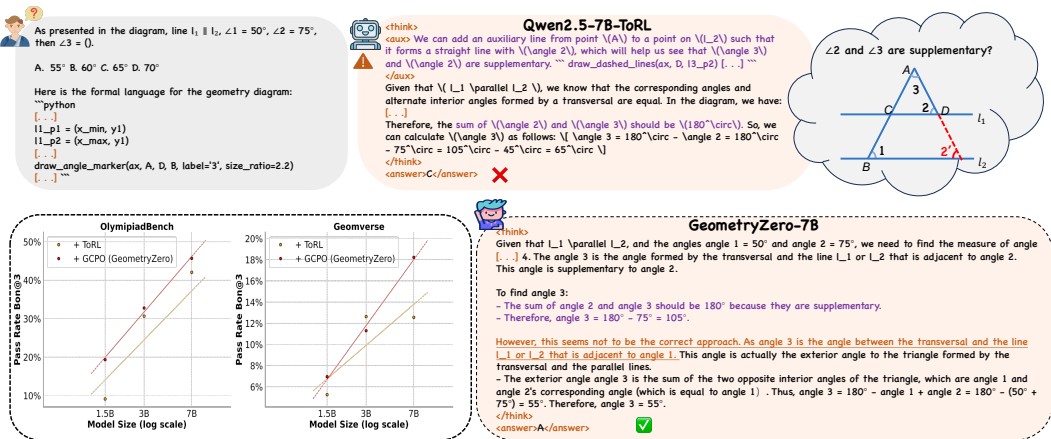

Figure 1: **A Comparative Study between ToRL and our GCPO. (a)** Two cases comparing GeometryZero-7B with Qwen2.5-7B-GRPO, revealing GeometryZero-7B judiciously determines to directly reason, while a ToRL-trained model indiscriminately conducts auxiliary construction. **(b)** Purple texts emphasize the erroneous reasoning steps both models undergo. The orange underlined texts amid reasoning process illustrate the critical reflection steps, which we identify as the model's *"aha moments"* (DeepSeek-AI et al., 2025) in geometric problem solving, from which GeometryZero-7B benefits in geometric problem solving scenarios. **(c)** GeometryZero showcases superior overall performance and better scaling effect across different model sizes compared to ToRL.

After the success of Deepseek-R1-Zero (DeepSeek-AI et al., 2025), GRPO has emerged as a generalizable and effective paradigm for both reasoning tasks and tool learning (Peng et al., 2025; Liu et al., 2025; Meng et al., 2025; Li et al., 2025). This makes it particularly suitable for training moderate-sized models capable of auxiliary construction while achieving a strong geometric reasoning performance. However, directly applying the GRPO framework to geometric reasoning with auxiliary construction presents challenges: in certain cases, forced or redundant auxiliary constructions prove unnecessary and potentially detrimental. Specifically, some problems can be solved through direct reasoning without auxiliary lines, where their forced inclusion may actually lead to incorrect solutions. Current RL approaches for tool use typically rely on unconditional rewards (consistently positive signals across all examples) to encourage indiscriminate tool invocation (Li et al., 2025). This approach lacks the flexibility to autonomously determine when auxiliary constructions are appropriate, thereby limiting RL's effectiveness in geometric problem solving.

We posit that a flexible mechanism is needed, allowing models to learn through RL when to use auxiliary construction and when to abstain. To this end, we propose **Group Contrastive Policy Optimization** (GCPO), a novel reinforcement learning approach that avoids the drawbacks of unconditional rewards. Specifically, GCPO differs crucially from traditional GRPO: it quantitatively estimates the benefits of auxiliary construction through two contrastive groups of rollouts, then provides flexible signals (conditional reward) including encouragements or penalties through Group Contrastive Masking. This mechanism enables GCPO to flexibly encourage auxiliary construction in clearly beneficial scenarios while punishing it in clearly detrimental situations. Inspired by LCPO (Aggarwal & Welleck, 2025), our work introduces a length reward to encourage multidimensional and more in-depth reasoning.

Building upon GCPO, we develop GeometryZero models, a series of lightweight (from 1.5B to 7B) LLMs specialized for geometric reasoning. Extensive experiments demonstrate that GeometryZero outperforms GRPO-trained models across multiple geometry problem-solving benchmarks, like Geometry3K and MathVista. As shown in Figure 1, by judiciously selecting scenarios for auxiliary construction rather than applying it indiscriminately, our GeometryZero showcases remarkable geometric reasoning and reflection ability, while achieving superior overall performance and better scaling across different model sizes compared to RL method with unconditional reward methods like ToRL (Li et al., 2025).

In summary, our contributions can be summarized as follows:

- We validate that through auxiliary construction during their reasoning process, LLMs can better solve complex tasks in geometric problem solving scenarios, where they utilize both contextual and altered formal languages for auxiliary construction.

- A novel reinforcement learning method called **GCPO** is proposed in our work, which flexibly provides either encouraging or punishing signals for auxiliary construction across different samples during reinforcement learning, avoiding models from indiscriminately applying auxiliary constructions while maintaining their benefits when strategically justified.

- We train GeometryZero models, a series of lightweight geometric reasoning models that judiciously determine when to employ auxiliary construction. We conduct extensive experiments on them and baselines, along with an in-depth ablation study on GCPO to validate the effectiveness of each component, plus detailed analyses revealing key insights about our approach.

## 2 RELATED WORK

### 2.1 GEOMETRY PROBLEM SOLVING

With the development of large language models (LLMs), researchers have begun to apply LLMs to geometric problem solving (Trinh et al., 2024). However, some early work such as Brehmer et al. (2023) primarily focuses on algebraic-level formal derivation, which has limited effectiveness in solving practical problems with numeric solutions. Other studies address the lack of geometry problem-solving data by proposing targeted benchmarks and datasets (Lu et al., 2021; Kazemi et al., 2023; Lu et al., 2024). Some recent work employs large-scale annotated data to perform supervised fine-tuning of models, aiming to enhance the performance of multimodal LLMs (Bi et al., 2024) on geometric problems (Gao et al., 2023).

Several approaches, such as GeoCoder (Sharma et al., 2024), attempt to utilize formal languages as context (e.g., code) to assist models in geometric reasoning. Other work explores the use of symbolic tools to strengthen models' geometric reasoning capabilities (Ning et al., 2025). Recent studies such as Hu et al. (2024); Wang et al. (2025); Chervonyi et al. (2025) propose encouraging models to construct auxiliary lines by modifying formal languages, thereby better leveraging the intrinsic properties of geometric context to solve the problems.

### 2.2 REINFORCEMENT LEARNING WITH VERIFIABLE REWARD

Reinforcement learning has long been a significant focus in the LLM research community (Schulman et al., 2017; Ouyang et al., 2022; Rafailov et al., 2024). Following the emergence of Deepseek-R1 (DeepSeek-AI et al., 2025), the research community has begun to focus on the application of reinforcement learning with verifiable reward, particularly GRPO (Shao et al., 2024), across various AI domains. Some studies attempt to reproduce GRPO's effectiveness in incentivizing reasoning capabilities on smaller LLMs (Peng et al., 2025). Others apply RLVR methods to multimodal LLMs to enhance their understanding of visual contexts (Meng et al., 2025; Liu et al., 2025). Additional work explores converting visual contexts into formal languages and utilizing reasoning LLMs for inference, aiming to surpass the capabilities of multimodal LLMs (Yang et al., 2025).

The GRPO algorithm was initially proposed in Shao et al. (2024) and applied to mathematical reasoning. Compared to PPO, it simplifies the reinforcement learning pipeline and eliminates the need for a critic model. CPPO (Lin et al., 2025) attempts to optimize the efficiency of the GRPO algorithm through pruning, reducing training costs while maintaining accuracy. DAPO (Yu et al., 2025) introduces a clipping mechanism and dynamic sampling to improve training diversity and stability. Liu et al. (2025) adapts GRPO's verifiable reward to visual perception tasks, enhancing model performance in visual reasoning. ToRL (Li et al., 2025) attempts to integrate tool-use rewards into GRPO, enhancing the model's tool invocation capability to improve its performance on mathematical reasoning. A separate work proposes a temporal reward coupled with accuracy reward to improve model grounding performance in video contexts (Feng et al., 2025).

## 3 PRELIMINARY

**Group Relative Policy Optimization (GRPO)** is a novel algorithm that leverages objectively verifiable supervision signals to enhance model performance on tasks requiring strong reasoning, such as mathematical and code-related problems. Compared with previous approaches, e.g., Reinforcement Learning from Human Feedback, which rely on trained reward models, GRPO utilizes direct verification functions to provide reliable reward feedback. This method simplifies the reward learning mechanism while enabling efficient alignment with the task's intrinsic correctness.

Specifically, given a question $\mathbf{q}$, the policy model $\pi_\theta$ generates a set of $N$ sampled outputs $\mathbf{O} = \{\mathbf{o}_1, \mathbf{o}_2, \ldots, \mathbf{o}_N\}$, where each output $\mathbf{o}_i$ receives a reward signal $\mathbf{r}_i$ through predefined verifiable reward functions. GRPO then optimizes the following clipped objective:

$$
\max_{\pi_\theta} \ \mathbb{E}_{\mathbf{O} \sim \pi_\theta(\mathbf{q})} \left[ \frac{1}{N} \sum_{i=1}^{N} \min \left( \frac{\pi_\theta(\mathbf{o}_i \mid \mathbf{q})}{\pi_{\theta_{\text{old}}}(\mathbf{o}_i \mid \mathbf{q})} \mathbf{A}_i, \ \text{clip} \left( \frac{\pi_\theta(\mathbf{o}_i \mid \mathbf{q})}{\pi_{\theta_{\text{old}}}(\mathbf{o}_i \mid \mathbf{q})}, \ 1 - \epsilon, \ 1 + \epsilon \right) \mathbf{A}_i \right) \right.
$$
$$
\left. - \beta \, \mathbb{D}_{\text{KL}} \left[ \pi_\theta(\mathbf{o} \mid \mathbf{q}) \, \| \, \pi_{\text{ref}}(\mathbf{o} \mid \mathbf{q}) \right] \right]. \tag{1}
$$

Here, $\pi_{\text{old}}$ denotes the policy before the current update, and $\pi_{\text{ref}}$ is a fixed reference policy (e.g., the initial model). $\mathbf{A}_i$ is the advantage estimate for output $\mathbf{o}_i$ based on its reward signal $\mathbf{r}_i$, $\epsilon$ is the clipping threshold, and $\beta$ is a hyperparameter controlling KL regularization to prevent excessive policy deviation.

Existing works, such as the DeepSeek R1-Zero (DeepSeek-AI et al., 2025) algorithm, abandon reliance on supervised fine-tuning and instead train entirely via reinforcement learning, particularly within the Group Relative Policy Optimization (GRPO) framework. In contrast to traditional reinforcement learning methods like PPO (Schulman et al., 2017), GRPO does not require a critic model to evaluate the policy's outputs. Given a question $q$, GRPO first generates $G$ distinct responses $\mathbf{O} = \{\mathbf{o}_1, \mathbf{o}_2, ..., \mathbf{o}_G\}$ using the current policy $\pi_{\theta_{\text{old}}}$. Then, the reward function is applied to obtain a set of verifiable rewards $\{\mathbb{R}(\mathbf{o}_i)\}$. By computing the mean and standard deviation of these rewards, GRPO normalizes them and estimates the advantage value for each response $\mathbf{o}_i$ as follows:

$$
\mathbf{A}_i = \frac{\mathbb{R}(\mathbf{o}_i) - \mathbb{E}_{\mathbf{o}_i \sim \mathbf{O}}[\mathbb{R}(\mathbf{o}_i)]}{\text{std}(\{\mathbb{R}(\mathbf{o}_i)\})}, \tag{2}
$$

where, $\mathbf{A}_i$ is the advantage value corresponding to the $i$-th response, representing its relative quality, $\mathbb{R}$ is the sum of verifiable rewards. The GRPO framework encourages the model to generate responses with higher verifiable rewards, thereby improving both reliability and correctness in reasoning-intensive tasks.

## 4 METHOD

### 4.1 OVERVIEW

**Group Contrastive Policy Optimization** (GCPO) introduces one key novel modification: it incorporates a crucial mechanism called group contrastive masking, which provides a positive mask for auxiliary reward in scenarios where auxiliary construction is beneficial, while applying a negative mask (i.e., penalty) in others. To achieve this objective, we propose the Group Contrastive Masking. GCPO also introduces an additional length reward optimized for longer completion, due to the requirement for auxiliary reasoning.

### 4.2 REINFORCEMENT LEARNING FOR AUXILIARY CONSTRUCTION

To enable models to incorporate auxiliary construction reasoning—a form of tool utilization (i.e. attempt to construct auxiliary lines in thinking process with formal language like tikz code)—we introduce an auxiliary reward that teaches the *"how-to"* capability, as in ToRL (Li et al., 2025), where an additional tool related reward is introduced for using coding for mathematical reasoning. During training, the textual context contains either TikZ code or logic form as detailed in Appendix A.1, which strictly corresponds to geometric diagrams, and the model is prompted to

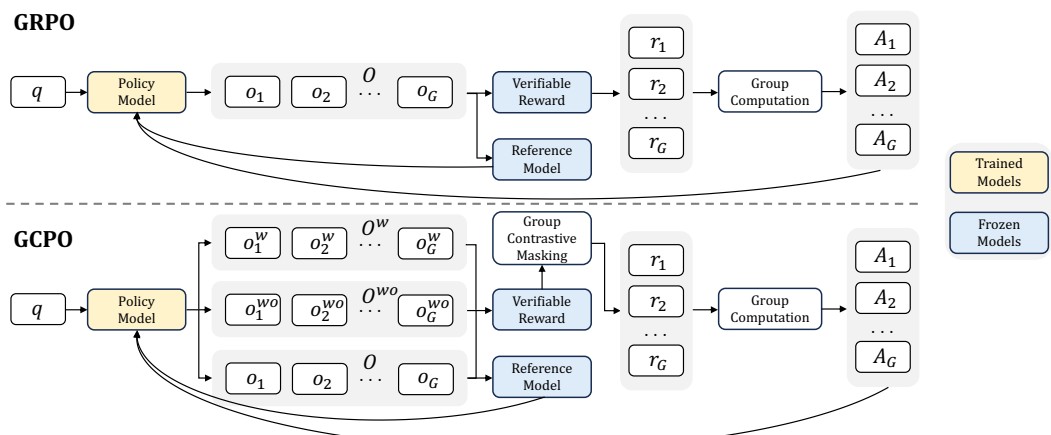

Figure 2: **The Illustration of GCPO.** One key difference between our GCPO and GRPO is Group Contrastive Masking: (1) GCPO samples two additional rollout groups $O^{\text{w}}$ and $O^{\text{wo}}$ for evaluating the quantitative benefits via *accuracy reward*. (2) The auxiliary reward signals of GCPO are dynamically masked to positive, negative, and zero during training as (Eq. 4). Another difference is that a novel length reward is also incorporated into the verifiable reward $\mathbb{R}$ during GCPO training.

autonomously decide whether to include auxiliary line construction in its reasoning process. For executable TikZ code, the auxiliary reward is positive if the altered tikz code in thinking process can execute successfully and render a diagram; for logic forms, we detect the presence of special tokens `<aux>` and `</aux>` indicating attempts to modify the logic form for auxiliary lines construction. Thus, the auxiliary reward for a given response $\mathbf{o}_i$ is defined as follow:

$$\mathbf{R}_{aux}(\mathbf{o}_i) = \begin{cases} 1 & \textit{if model constructs auxiliary lines}, \\ 0 & \textit{otherwise}. \end{cases} \tag{3}$$

### 4.3 Conditional Reward for Auxiliary Construction

Inspired by existing works (Chervonyi et al., 2025; Hu et al., 2024), we aim to endow models with the capability of auxiliary construction with formal language in geometric problem solving. However, it is crucial to note that while teaching models *"how to"* is important (Li et al., 2025), we must also teach them *"when to do it"*. Although GRPO has proven effective in enhancing reasoning capabilities, it lacks conditional rewards for tool usage and relies solely on unconditional rewards to encourage desired behaviors, which may cause indiscriminate use of the tool in certain scenarios. To address this limitation, we propose Group Contrastive Policy Optimization (**GCPO**), which introduces a group contrastive reward mechanism for a conditional reward signal that flexibly provides encouragements or penalties during training, allowing the flexibility of the trained models of whether to apply the tool (i.e. auxiliary construction) or not.

The key insight of GCPO stems from the observation that in geometric problem solving, while auxiliary lines can enhance reasoning in many cases, they may be unnecessary or even detrimental in some cases as well. Unconditional encouragement of auxiliary construction could lead to sub-optimal performance for complex scenarios like geometry problems, thus, we need to incorporate a conditional reward during reinforcement learning.

### 4.4 Components of Group Contrastive Policy Optimization

**Group Contrastive Masking** The core idea of GCPO is that during training, models should become aware of the quantifiable benefits of using code to draw auxiliary lines - employing this capability when beneficial and avoiding it when counterproductive. As shown in Figure 2, during response sampling, the model generates $G$ rollouts $\mathbf{O} = \{\mathbf{o}_1, \mathbf{o}_2, \cdots, \mathbf{o}_G\}$, along with another two contrastive rollout groups: one requiring auxiliary line thought process group $\mathbf{O}^{\text{w}} = \{\mathbf{o}_i^{\text{w}}\}$ and one

group prohibiting it $\mathbf{O}^{\text{wo}} = \{\mathbf{o}_i^{\text{wo}}\}$. The group contrastive masking function is defined as:

$$\mathbf{Mask}(\mathbf{R}_{aux}(\mathbf{O})) = \begin{cases} \mathbf{R}_{aux}(\mathbf{O}) & \textit{if } \mathrm{E}(\mathbf{R}_{acc}(\mathbf{O}^{\text{w}})) > \mathrm{E}(\mathbf{R}_{acc}(\mathbf{O}^{\text{wo}})) + \epsilon, \\ -\mathbf{R}_{aux}(\mathbf{O}) & \textit{if } \mathrm{E}(\mathbf{R}_{acc}(\mathbf{O}^{\text{wo}})) > \mathrm{E}(\mathbf{R}_{acc}(\mathbf{O}^{\text{w}})) + \epsilon, \\ \mathbf{0} & \textit{otherwise}, \end{cases} \tag{4}$$

where $\mathbf{R}_{acc}(\mathbf{o}_i)$ represents the accuracy reward function indicating whether response $\mathbf{o}_i$ contains the correct final answer, $\mathbf{R}_{aux}(\mathbf{o}_i)$ refers to the auxiliary reward in (Eq. 3) and $\epsilon$ (set to 0.05 in experiments) is a threshold hyperparameter controlling reward masking. Following standard mathematical conventions, the function $\mathbf{R}_{aux}$ can naturally extend from a single response to a set of responses: $\mathbf{R}_{aux}(\mathbf{O}) = \{\mathbf{R}_{aux}(\mathbf{o}_1), \ldots, \mathbf{R}_{aux}(\mathbf{o}_G)\}$, which also holds for $\mathbf{R}_{acc}(\mathbf{O}^{\text{w}})$ and $\mathbf{R}_{acc}(\mathbf{O}^{\text{wo}})$.

**Length Reward** Auxiliary construction thinking requires deeper, multi-dimensional analysis, necessitating longer reasoning processes. Inspired by Length Controlled Policy Optimization (LCPO) (Aggarwal & Welleck, 2025), we adapt by introducing a simplified length reward, where $\mathrm{len}(\mathbf{o}_i)$ counts tokens in response $\mathbf{o}_i$ and $l_{\max}$ is the maximum allowed completion length.

$$\mathbf{R}_{length}(\mathbf{o}_i) = \min\{1, \frac{\mathrm{len}(\mathbf{o}_i)}{l_{\max}}\} \tag{5}$$

**Verifiable Reward Combination** As a variant of RLVR, the verifiable reward $\mathbb{R}(\mathbf{o}_i)$ of GCPO combines multiple weighted components as below, where $\mathbf{R}_{\text{GRPO}}(\mathbf{o}_i)$ includes a accuracy reward and a format reward ensuring proper output structure, while hyperparameter $\lambda$ representing auxiliary reward weight and hyperparameter $\beta$ representing length reward weight are both set to 0.5.

$$\mathbb{R}(\mathbf{o}_i) = \mathbf{R}_{\text{GRPO}}(\mathbf{o}_i) + \lambda \cdot \mathbf{Mask}(\mathbf{R}_{aux}(\mathbf{o}_i)) + \beta \cdot \mathbf{R}_{length}(\mathbf{o}_i) \tag{6}$$

In essence, GCPO uses masked auxiliary rewards to teach appropriate tool usage contexts, while length rewards ensure sufficient reasoning capacity. The framework largely inherits GRPO's verifiable reward framework, employing outcome-based reinforcement learning for model training.

## 5 EXPERIMENTS

### 5.1 EXPERIMENTAL SETTING

For dataset, we apply a synthesized dataset with data sampled from two popular geometry problem solving (GPS) dataset including Geomverse (Kazemi et al., 2023) and Geometry3k (Lu et al., 2021), for training SFT and RL models. The training dataset recipe is detailed is Appendix A.1. For training details, we utilize the vLLM inference framework (Kwon et al., 2023) during training and evaluation. More details are presented in Appendix A.2.

As for models, inspired by (Yang et al., 2025), we turn the visual diagrams into formal language contexts for better reasoning performance. Thus, we use Qwen2.5 (Qwen et al., 2025) series language models for training. For benchmarks, besides the in-domain benchmarks of Geometry3k and Geomverse, the OOD geometry benchmarks comprise MathVista (Lu et al., 2024) and Olympiad-Bench (He et al., 2024). The details are provided in Appendix A.3.

**Baselines** To fully demonstrate the effectiveness of **GCPO**, we compare against the following baselines: **(1)** SFT. The model undergoes supervised fine-tuning using prompt-response pairs, where responses are either human-annotated or distilled from capable LLMs. **(2)** GRPO (Shao et al., 2024). A reinforcement learning via verifiable outcome algorithm where the model generates multiple responses for baseline advantage estimation to encourage reasoning and improve problem-solving, which eliminates the usage of a critic model or reward model. **(3)** ToRL (Li et al., 2025). An RL algorithm building on GRPO that appends an additional reward function (Eq. 3) to unconditionally encourage tool usage (i.e., auxiliary construction).

### 5.2 RESULTS

As shown in the table 1, our experimental results across four geometry problem-solving benchmarks demonstrate GCPO's effectiveness in enhancing model capabilities on geometric problems. Key findings are summarized below:

| Model Size | Method | Geomverse | Geometry3k | MathVista | OlympiadBench | Avg. |
|---|---|---|---|---|---|---|
| | Qwen2.5-1.5B-Instruct | 4.20 | 41.76 | 47.70 | 13.44 | 26.78 |
| | + SFT | 4.80 | 44.25 | 43.11 | 14.51 | 26.67 |
| 1.5B | + GRPO | 5.76 | 53.35 | 57.79 | 14.51 | 32.85 |
| | + ToRL | 5.26 | 57.01 | 57.79 | 11.29 | 32.84 |
| | **GeometryZero-1.5B** (*ours*) | **6.96** | **60.23** | **61.77** | **19.35** | **37.08** |
| | Qwen2.5-3B-Instruct | 10.53 | 65.83 | 67.88 | 32.25 | 44.12 |
| | + SFT | 10.20 | 71.65 | 73.08 | 30.64 | 46.39 |
| 3B | + GRPO | 12.13 | 75.87 | **82.87** | 31.72 | 50.65 |
| | + ToRL | **12.63** | 77.31 | 81.34 | 33.87 | 51.29 |
| | **GeometryZero-3B** (*ours*) | 11.30 | **79.25** | 82.56 | **35.48** | **52.15** |
| | G-Llava-7B | 6.23 | 49.31 | 46.92 | 27.82 | 32.57 |
| | GNS-Llava-1.5-7B | 5.21 | 62.00 | 51.40 | 33.54 | 38.04 |
| | Qwen2.5-7B-Instruct | 14.76 | 70.99 | 68.19 | 39.24 | 48.30 |
| 7B | + SFT | 15.36 | 75.98 | 76.14 | 41.93 | 52.35 |
| | + GRPO | 16.93 | **79.03** | 86.23 | 40.32 | 55.63 |
| | + ToRL | 12.56 | 78.75 | 83.48 | 44.08 | 54.72 |
| | **GeometryZero-7B** (*ours*) | **18.23** | 78.81 | **87.15** | **45.69** | **57.47** |

Table 1: **The main empirical results.** The BoN@3 pass rate results across in-domain benchmarks including Geomverse, Geometry3k and out-of-domain results on MathVista and OlympiadBench, where the best results are **bold**. Results from our GeometryZero (w.r.t., + GCPO) models are shown in gray part.

**SFT Memorizes while RL Generalizes.** We observe that SFT models (Qwen2.5-1.5B-SFT and Qwen2.5-3B-SFT) show consistent improvements over original Instruct models on in-domain benchmarks like Geomverse and Geometry3k. For instance, Qwen2.5-1.5B-SFT and Qwen2.5-3B-SFT gains an improvement of 2.49% and 5.83% on Geometry3k. However, these SFT models exhibit either performance drops or smaller gains compared to RL methods on OOD benchmarks like MathVista and OlympiadBench. For instance, while Qwen2.5-1.5B-SFT exhibits a performance decline of 4.59% compared to the base model on the OOD benchmark MathVista, Qwen2.5-1.5B-GRPO demonstrates a notable improvement of 10.09%. Overall, RL approaches including GRPO, ToRL, and GCPO achieve more consistent improvements across both in-domain and OOD benchmarks, surpassing SFT and proving the effectiveness of reinforcement learning.

**Group Contrastive Policy Optimization Works.** Compared to GRPO, ToRL models unconditionally encourage auxiliary construction during reasoning process across all examples with an unconditional reward design (Eq. 3). The empirical results demonstrate that ToRL models has no clear advantage over GRPO across various model scales, indicating that this coarse-grained policy fails to provide significant benefits for auxiliary construction in geometric problem-solving scenarios. For instance, while ToRL demonstrates a marginal 0.64% advantage over GRPO on 3B models, it exhibits a 0.91% performance reduction on 7B models. In contrast, GCPO improves model performance on both in-domain and OOD benchmarks, achieving consistently better average performance on most benchmarks across model sizes. This indicates discerning when to incorporate auxiliary reasoning ultimately improving problem-solving capabilities. As shown in Figure 3, GCPO enhances geometric problem-solving by generating auxiliary constructions during the reasoning process, we also provide more case studies in Appendix G.

## 5.3 ABLATION STUDY

| Base Model | Method | AR | GC | LR | Geomverse | Geometry3k | MathVista | OlympiadBench | Avg. |
|---|---|---|---|---|---|---|---|---|---|
| | GRPO | ✗ | ✗ | ✗ | 16.93 | 79.03 | 86.23 | 40.32 | 55.63 |
| | GCPO (/wo AR) | ✗ | ✗ | ✓ | 17.40 | 79.53 | 87.15 | 41.93 | 56.50 |
| Qwen2.5-7B-Instruct | GCPO (/wo LR, /wo GC) | ✓ | ✗ | ✗ | 12.56 | 78.75 | 83.48 | 44.08 | 54.72 |
| | GCPO (/wo LR) | ✓ | ✓ | ✗ | **18.59** | **80.86** | 86.54 | 41.93 | 56.98 |
| | **GeometryZero** | ✓ | ✓ | ✓ | 18.23 | 78.81 | **87.15** | **45.69** | **57.47** |

Table 2: **The ablation study of GCPO components**. The components includes auxiliary reward (AR), group contrastive (GC) masking and length reward (LR).

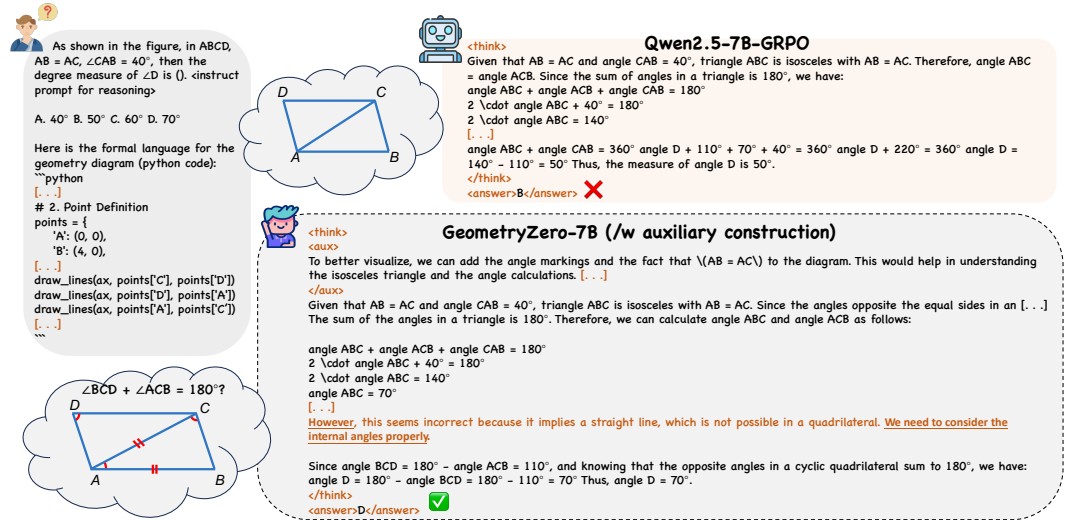

Figure 3: **A Case Study between GRPO and GCPO.** Two reponses comparing Qwen2.5-7B-GRPO with our GeometryZero-7B for a MathVista problem, revealing how GeometryZero-7B effectively constructs auxiliary elements during its reasoning process. The orange underlined texts during reasoning process are reflection process in geometric problem solving.

To better understand the contributions of components in GCPO, we conduct an ablation study to evaluate three variants of GeometryZero and compare them with the GRPO model and GeometryZero, where the descriptions of the variants are further detailed in Appendix C.1.

Our findings show that GeometryZero (/wo LR) achieves on average 2.26% higher performance than GeometryZero (/wo LR, /wo GC). Both GeometryZero (/wo AR) and GeometryZero (/wo LR) demonstrate better average performance across benchmarks compared to Qwen-2.5-7B-GRPO by 0.87% and 1.35% respectively, while these two variants show 0.97% and 0.49% lower average performance than GeometryZero. We also provide an ablation study on 3B models in Appendix C.2.

The experimental results indicate that removing either the auxiliary reward or its corresponding group contrastive masking leads to performance degradation across benchmarks. Similarly, eliminating the length reward in GCPO also poses negative effects. These results validate the effectiveness of our proposed method.

# 6 DISCUSSION

## 6.1 COMPLETION LENGTH OF MODELS

Response length serves as a crucial metric for observing training dynamics in RL (Meng et al., 2025). We monitor the variation in response length during the training process of GeometryZero and GRPO models as shown in Figure 4. For 7B models, we observe the following trends in response length: During the initial few steps, the model's response length increases rapidly, subsequently it decreases, after reaching the lowest point, it then begins to rise again.

The phenomenon aligns with observations in llm-r1 (Peng et al., 2025). We hypothesize that in the first phase, the model is encouraged by format rewards to learn reasoning patterns that generate thoughts before answers, leading to increased output length. In the second phase, as training progresses, the model begins to optimize the reward function, particularly the accuracy reward, causing it to reduce redundant outputs while maintaining the required format, resulting in decreased response length. For the third phase, we speculate that in later training stages, the model learns more sophisticated reasoning patterns and attempts to generate more complex reasoning steps, leading to the length recovery.

The observation differs for 1.5B models. GeometryZero-1.5B exhibits the rise-fall-rise pattern in response length, while the GRPO model shows no recovery in response length in the last stage.

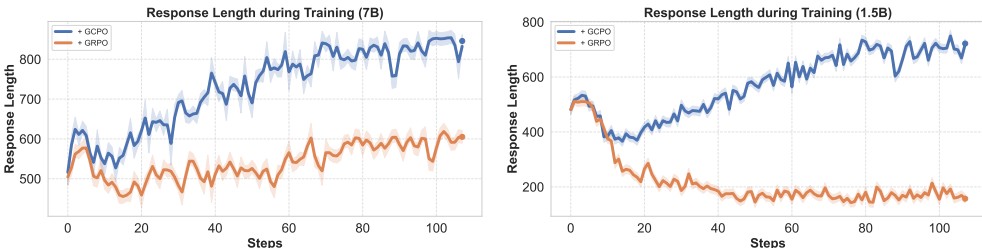

Figure 4: (LEFT) The trend of **completion length** during reinforcement learning of 7B models. (RIGHT) The trend of **completion length** during reinforcement learning of 1.5B models. **We observe that the completion length of GCPO models follows a distinct pattern during training: initially increasing, then decreasing, before rising again, which could also be observed for 7B GRPO models.**

We attribute this to the model's limited capacity due to smaller parameter size, which prevents it from learning more comprehensive and profound reasoning processes through GRPO alone in later training stages.

## 6.2 MASK RATIO OF GCPO

According to (Eq. 4), our method's characteristic is that during Group Masking, it applies positive masks to auxiliary rewards for some cases, negative masks to others, while zero-masking cases where the mean accuracy reward gap does not exceed epsilon. As presented in Figure 6, we observe that while the overall proportions of positive and negative masks fluctuate, they remain generally stable during training, with positive masks consistently outnumbering negative masks.

This phenomenon demonstrates that the rollout group with auxiliary construction (i.e. $O^{\mathrm{w}}$) achieves higher accuracy rewards than the group without auxiliary construction (i.e. $O^{\mathrm{wo}}$) in reward score computing, indicating that auxiliary construction generally contributes to obtaining correct solutions and thus validating its effectiveness. More records of group mask ratio are presented in appendix H.4.

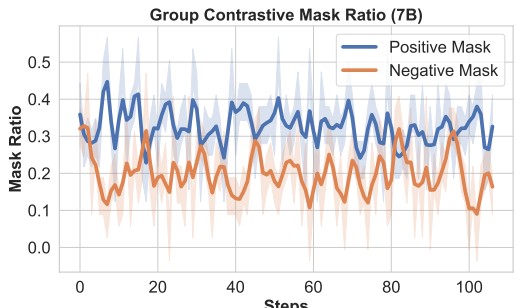

Figure 5: The positive group mask and negative group mask ratio in group contrastive masking (Eq. 4). We consider Mask Ratio as an important metric for observing GCPO training dynamics, as it represents the proportion of cases deemed either "auxiliary construction is useful" or "auxiliary construction is harmful" during training.

We also demonstrate more in-depth discussions for epsilon settings in Appendix B and the performance of GeometryZero models on geometry proving tasks in Appendix D.

## 7 CONCLUSION

In this paper, we propose **Group Contrastive Policy Optimization**, a novel reinforcement learning framework that incorporates verifiable rewards to optimize conditional reward particularly for auxiliary construction in geometric reasoning. GCPO dynamically adapts to different problem scenarios, supporting an autonomous strategy of tool-assisted and tool-free reasoning. Building upon this framework, we introduce GeometryZero, a series of geometric reasoning models that autonomously learn when and how to apply auxiliary constructions during the reasoning process. Extensive experiments demonstrate the effectiveness of our approach, while detailed analyses provide insights for future research directions.

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

# A    IMPLEMENTATION DETAILS

## A.1    TRAINING DATASET CONSTRUCTION

| Dataset | Sample Size | Code Type | Code Executable |
|---------|-------------|-----------|-----------------|
| **Geomverse** | $2k$ | *Tikz Code* | ✓ |
| **Geometry3k** | 1443 | *Logic Form* | ✗ |

Table 3: **The training dataset construction details.** The training data are sampled from two popular geometry problem solving (GPS) dataset including Geomverse and Geometry3k.

To ensure the model adequately learns geometric problem solving, we select two mainstream geometric problem solving (GPS) datasets. Our training data comes from Geometry3k (Lu et al., 2021) and Geomverse (Kazemi et al., 2023).

- Geometry3k. We randomly select 1443 training samples from Geometry3k. For the SFT experiments, this dataset lacks supervised sequences, so we use Qwen2.5-72B-Instruct (Qwen et al., 2025) to generate CoT reasoning processes with known answers. These reasoning processes are concatenated with the solutions to form supervised responses. For RL-based methods like GRPO and GCPO, we only utilize the problems in the dataset and employ the final answers as supervision.

- Geomverse. We randomly choose $2k$ training samples from Geomverse. Since this dataset already contains human-annotated CoT processes, we directly use them for SFT experiments. We also only employ the problems in the dataset and the final answers as supervision for RL-based methods.

## A.2    TRAINING DETAILS

We set train batch size to 32 and micro train batch size to 1, for response sampling we apply a rollout batch size of 64 and a micro rollout batch size of 2. We set max prompt length to 2048 and max completion length $l_{max}$ to 1024. We use full parameter tuning rather than PEFT methods (Bi et al., 2025).

We set $G$ to 8, with both the SFT learning rate and the GRPO learning rate at $3e-7$ and the format reward weight set to 0.5. Due to the limited training data and absence of significant policy shift concerns, we set the KL coefficient to 0 to achieve better tuning performance. As for compute hardware, we use 4 Nvidia H100 GPUs for training and later evaluation.

## A.3    EVALUATION BENCHMARKS

To comprehensively evaluate the model's performance on geometric problem solving, we conduct evaluations on several mainstream geometric problem benchmarks. Besides using Geometry3k and the Geomverse D2 subset to test the model's in-domain geometric capabilities, for out-of-distribution problems, we also evaluate the model's performance on MathVista and Olympiad-Bench.

Besides the in-domain benchmarks, the OOD geometry benchmarks comprise:

- MathVista (Lu et al., 2024). A consolidated mathematical reasoning benchmark within visual contexts. To evaluate LLMs on geometric problems, GPT-4o converts visual contexts from MathVista testmini into textual Python code using ReACT and Self-Vote mechanisms. We then manually verify that the code-generated graphics match the original visual contexts, resulting in an evaluation set containing 109 samples.

- OlympiadBench (He et al., 2024). The benchmark is an Olympiad-level multimodal scientific benchmark. We extract all geometry problems and filter for those with only one solution to ensure single-solution supervision. Using the same pipeline as MathVista, we convert visual contexts into LLM-comprehensible Python code, obtaining an evaluation set of 62 samples to assess model performance on Olympiad-level geometry problems.

## B  THE IMPACT OF HYPERPARAMETER $\epsilon$ OF GROUP CONTRASTIVE MASKING

To provide more insightful analysis of our method, we conduct a comparative study with different epsilon hyperparameter settings. We set epsilon values at 0, 0.05, 0.15, 0.3, and 1.0 separately for training GeometryZero and evaluating their benchmark performance. As presented in Figure 6, we find that as epsilon increases from 0 to 1.0, the algorithm's performance first improves slightly and then declines.

We speculate that when epsilon is too low, the algorithm applies positive or negative masks to cases where the benefit of auxiliary construction is uncertain, leading to unstable training in these cases and ultimately affecting model performance. When epsilon is too high, the

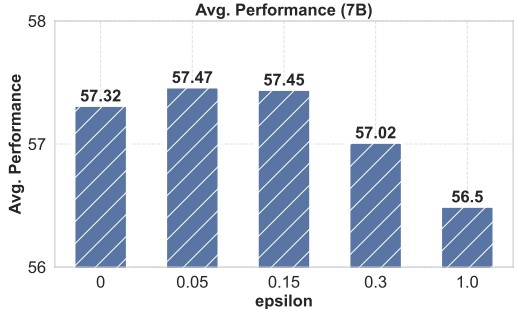

Figure 6: The average performance of GeometryZero with different hyperparameter epsilon settings in GCPO training.

threshold for group contrastive masking becomes excessively strict, causing auxiliary rewards to be zero in most cases, which effectively renders the auxiliary reward mechanism inoperative. We conclude that GCPO performs best in the epsilon range of 0.05 to 0.15, and thus we keep epsilon at 0.05 in our experiments.

## C  ABLATION STUDY

### C.1  VARIANT MODELS IN ABLATION STUDY

Here are the model variants used in ablation study, serving as a supplementary material for section 5.3:

- GeometryZero (/wo AR), which excludes the auxiliary construction reward (Eq. 3) and consequently removes the Group Contrastive Masking mechanism (Eq. 4), retaining only the length penalty term (Eq. 5);

- GeometryZero (/wo LR, /wo GC), which only retains the auxiliary reward (Eq. 3) encouraging auxiliary construction thinking during the reasoning phase but excludes the corresponding Group Contrastive Masking (Eq. 4), equivalent to ToRL using unconditional auxiliary reward;

- GeometryZero (/wo LR), which excludes the length reward (Eq. 5) in GCPO that encourages longer reasoning chains, retaining other components of GCPO.

### C.2  ABLATION STUDY ON 3B MODEL

| Base Model | Method | AR | GC | LR | Geomverse | Geometry3k | MathVista | OlympiadBench | Avg. |
|---|---|---|---|---|---|---|---|---|---|
| | GRPO | ✗ | ✗ | ✗ | 12.13 | 75.87 | **82.87** | 31.72 | 50.65 |
| | GCPO (/wo AR) | ✗ | ✗ | ✓ | 12.60 | 75.20 | 81.65 | 33.87 | 50.83 |
| Qwen2.5-3B-Instruct | GCPO (/wo LR, /wo GC) | ✓ | ✗ | ✗ | 12.63 | 76.37 | 81.34 | 33.87 | 51.05 |
| | GCPO (/wo LR) | ✓ | ✓ | ✗ | **12.90** | 78.20 | 81.65 | 32.25 | 51.25 |
| | **GeometryZero** | ✓ | ✓ | ✓ | 11.30 | **79.25** | 82.56 | **35.48** | **52.15** |

Table 4: **The ablation study of GCPO components on Qwen2.5-3B-Instruct**. The components includes auxiliary reward (AR), group contrastive (GC) masking and length reward (LR).

## D  GEOMETRYZERO ON GEOMETRIC PROVING TASKS

In widely used geometry benchmarks, UniGeo (Chen et al., 2022) contains a subset of geometric proof problems. For efficient comparison, we selected 108 problems of this subset for our additional experiments.

Since AlphaGeometry Trinh et al. (2024) requires a strict geometric `DSL` (formal language describing points, lines, circles, relations), we first used GPT-4o to batch-formalize the 108 UniGeo problems into `DSL`. The correctness of the proofs was then verified using an automated validation script. For GPT-4o and GeometryZero, we generated complete proof sequences and compared them with golden sequences to measure accuracy on proof problems.

AG's primary bottleneck lies in formalizing problems into `DSL`, which accounts for the imperfection of its accuracy. Actually, the difficulty of UniGeo problems does not necessitate AG's symbolic search process. GeometryZero-14B and GPT-4o achieve comparable performance, with GeometryZero-14B showing a 7.4% improvement over Qwen2.5-14B-Instruct, despite the absence of proof problems in its training data. This highlights the strong generalization capability of GCPO.

| Model | UniGeo (proof part) |
|---|---|
| AlphaGeometry (Trinh et al., 2024) | 94.4% |
| GPT-4o | 74.1% |
| Qwen2.5-14B-Instruct | 64.8% |
| GeometryZero-14B | 72.2% |

Table 5: The performance of different models including AG (Trinh et al., 2024) and GeometryZero-14B on UniGeo (geometric proof part).

## E  LIMITATIONS

While GCPO demonstrates strong performance, several limitations warrant discussion. First, our method assumes access to verifiable reward signals, which may not be available for all geometry problem types (e.g., inductive geometric proof). Second, the approach requires careful hyperparameter tuning for the contrastive rewards, suggesting a need for more robust automated configuration methods. Additionally, due to compute constraints, we limited our experiments to moderate model sizes (under 7B parameters). These limitations point to valuable directions for future research in reasoning systems for geometric problems.

## F  DECLARATION ON LLM USAGE

In this paper, we use large language models (e.g., GPT-5) solely for minor language polishing. The models were not used to generate ideas, analyze data, write code, or conduct experiments. **All scientific claims, analyses, and conclusions are the authors' own**; all edits were reviewed by the authors, who accept full responsibility for any remaining errors.

## G  CASE STUDY

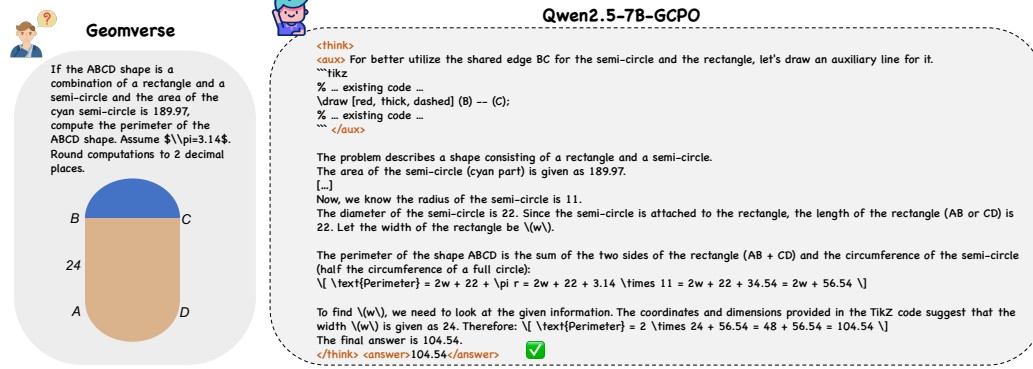

Figure 7: A case example from Geomverse Kazemi et al. (2023) of GeometryZero-7B (Qwen2.5-7B-GCPO), amid the reasoning process the model outputs executable tikz code to construct auxiliary lines for geometric reasoning.

# H TRAINING DYNAMICS DURING REINFORCEMENT LEARNING

## H.1 ACCURACY REWARD

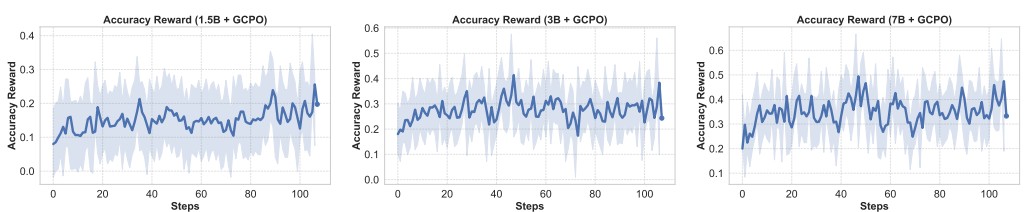

Figure 8: The trend of accuracy reward of **GeometryZero** (GCPO) models during training.

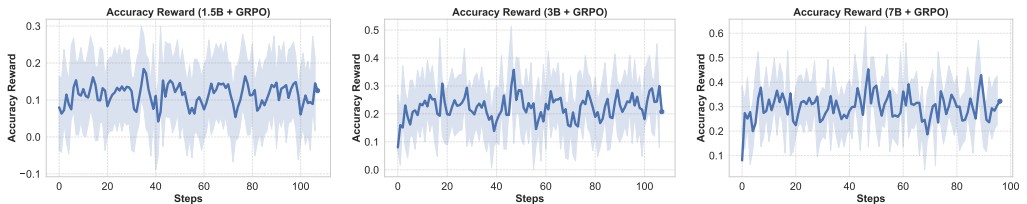

Figure 9: The trend of accuracy reward of GRPO models during training.

## H.2 FORMAT REWARD

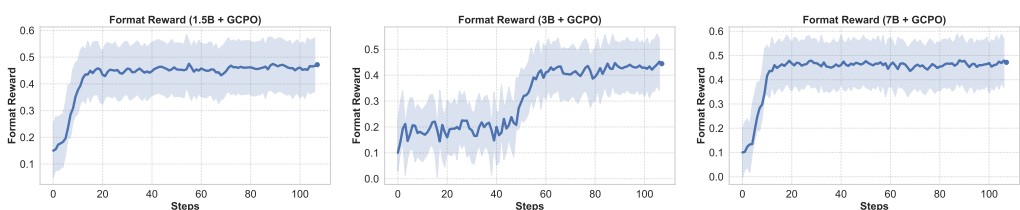

Figure 10: The trend of format reward of **GeometryZero** (GCPO) models during training.

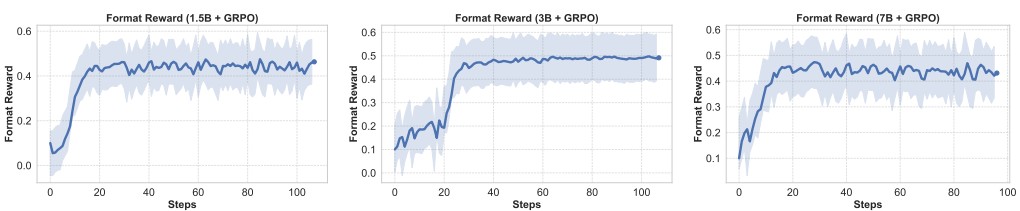

Figure 11: The trend of format reward of GRPO models during training.

## H.3 COMPLETION LENGTH

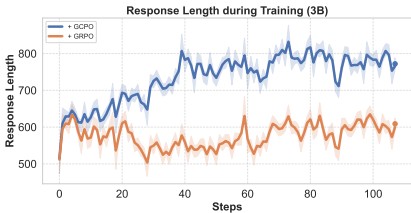

Figure 12: **The trend of response length of GCPO and GRPO during training on 3B models.** For 3B models, We also observe the completion length of follows a distinct pattern during training: initially increasing, then decreasing or stagnating, before rising again.

## H.4 MASK RATIO

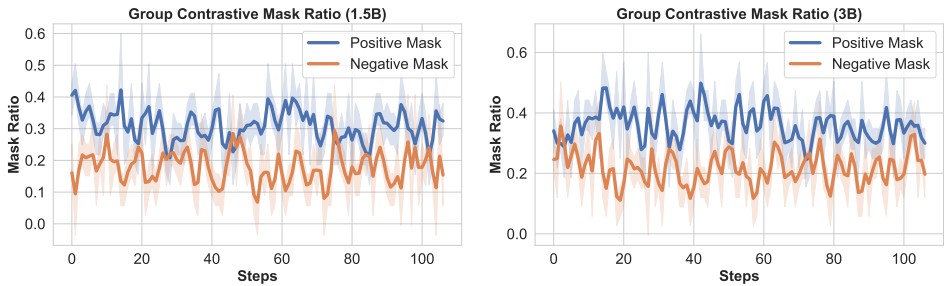

Figure 13: **The record of group mask ratio.** The positive group mask and negative group mask ratio in group contrastive masking for 1.5B and 3B models.

