# OpenReview forum: "GeometryZero: Advancing LLM Geometry Solving via Group Contrastive Policy Optimization"
_ICLR.cc/2026/Conference — ICLR 2026 Conference Withdrawn Submission_

### Official Review · Reviewer_LeaT · 2025-10-16

**Soundness:** 2
**Presentation:** 3
**Contribution:** 1
**Rating:** 2
**Confidence:** 5

**Summary:**

This paper introduces Group Contrastive Policy Optimization (GCPO), a reinforcement learning framework designed to improve geometric problem solving by selectively encouraging auxiliary line construction. GCPO differs from prior methods by applying conditional rewards based on the effectiveness of auxiliary constructions, using a group contrastive masking mechanism. It also adds a length reward to support longer, multi-step reasoning processes. Built on GCPO, the authors train GeometryZero, a set of small-scale models that achieve improved accuracy without relying on large-scale LLMs. The approach is evaluated on several benchmarks, showing consistent gains over existing RL baselines.

**Strengths:**

1. The paper tackles the often-overlooked question of whether to use auxiliary lines, not just how, by introducing a contrastive reward mechanism that encourages or discourages use based on actual benefit.

2. The authors run clear ablations across model sizes, showing how each component (auxiliary reward, masking, length reward) impacts performance. And they provide useful insights into training dynamics, like how response length and masking evolve.

**Weaknesses:**

1. The optimization goal is overly narrow. The method focuses only on auxiliary line construction, which is just one of many challenges in geometric reasoning; other strategies such as setting up coordinate systems or using reverse reasoning are not discussed.

2. The reward estimation is inefficient and potentially inaccurate. Using the accuracy gap between two rollout groups to decide whether to apply auxiliary lines is both costly and noisy; it would be more reliable to pre-annotate which problems benefit from auxiliary constructions or design a list-wise reward based on multiple rollouts.

3. The methodological novelty is limited. The work mainly introduces reward engineering on top of GRPO rather than proposing a new algorithm.

4. The experiments are not fully convincing. The models are only trained on the Qwen2.5 family, and the chosen baselines such as GNS-LLaVA are outdated; the paper lacks comparison with newer geometry-specific or multimodal reasoning models.

5. The experimental analysis is insufficient. Although the model improves on out-of-domain benchmarks like MathVista and OlympiadBench, there is no detailed analysis explaining which kinds of problems contributed to these gains.

6. The study only covers text-based geometry problems. It remains unclear how the proposed method would perform with real visual inputs or under multimodal settings.

**Questions:**

1. On the OOD benchmarks MathVista and OlympiadBench, which types of problems show the largest improvement? The paper only presents in-domain case studies but lacks OOD examples.

2. How does GCPO perform compared to stronger multimodal SOTA models such as GPT‑4V or Gemini? Can the method be transferred to such visual‑language models?

3. Since the training data come only from Geometry3k and Geomverse (~3.4k samples), will GCPO’s advantage still hold on larger or more diverse datasets? Are there plans to test its generalization further?

4. Is auxiliary line construction truly the core difficulty in geometric reasoning? Has there been any analysis of other potential reasoning bottlenecks in geometry problem solving?

---

### Official Review · Reviewer_QdLU · 2025-11-01

**Soundness:** 3
**Presentation:** 3
**Contribution:** 2
**Rating:** 4
**Confidence:** 5

**Summary:**

This paper proposes Group Contrastive Policy Optimization (GCPO), a new reinforcement learning algorithm designed to improve LLM-based geometric problem solving, especially by enabling judicious use of auxiliary constructions (such as drawing extra geometric lines) in reasoning. GCPO introduces a group contrastive masking mechanism providing context-sensitive rewards (encouragement or penalty) for auxiliary construction, and augments this with a length reward to promote deeper reasoning. Building on GCPO, the authors present GeometryZero, a family of lightweight models that surpass standard RL (GRPO, ToRL) and SFT baselines across multiple geometry benchmarks, as validated by extensive experiments (Geometry3K, Geomverse, MathVista, and OlympiadBench) and ablation studies.

**Strengths:**

1. Motivated, Well-Positioned Task: The focus on geometry problem-solving with LLMs is timely and important, given the unique challenges of integrating formal diagram understanding and reasoning.

2. Methodological Innovation: GCPO’s group contrastive masking is a significant step forward in RL for tool-augmented reasoning, dynamically providing positive/negative reward signals depending on the utility of auxiliary constructions. This is a tangible conceptual and algorithmic advance beyond unconditional reward methods like ToRL.

3. Accessible and Well-Documented: Implementation details, dataset construction (Table 3), and training pipeline are presented in sufficient detail for reproduction; the appendix is extensive, supporting the main claims.

**Weaknesses:**

1. Empirical Scope Limitation: All experiments are restricted to models under 7B parameters, whereas recent trends in geometric reasoning often report results for larger, near-state-of-the-art models. This limits claims for broader applicability; for instance, Table 1 lacks comparison to the largest scale public models (e.g., GPT-4o, Gemini), even if computational restrictions are real.

2. Definitional Ambiguity in Conditional Masking: In Equation 4 and its surrounding narrative (Pages 5–6), the treatment of $\epsilon$ is reasonable, but the masking decision relies on mean accuracy reward differentials; this could underplay per-instance variability or rare-case utility/harm of auxiliary construction (i.e., GCPO may still reinforce suboptimal construction in edge cases). More discussion or empirical analysis of these edge cases, perhaps via qualitative categorization, would help.

3. Regarding Auxiliary Lines and Answer Length: Intuitively, the construction of auxiliary lines and increased answer length should contribute to solving more challenging geometry problems. Although experimental results indicate quantitative improvements on existing benchmarks, more concrete examples (such as visualizations) are necessary to convincingly demonstrate this enhancement for high-difficulty tasks.

4. Regarding the Use of TikZ: The use of TikZ code for generating auxiliary lines requires further clarification. During reasoning, does the system construct a new image to perform interleaved visual and textual inference? Furthermore, TikZ might not be the optimal tool for drawing auxiliary lines, primarily due to potential precision issues.

**Questions:**

please refer to the weaknesses

---

### Official Review · Reviewer_ewCt · 2025-11-01

**Soundness:** 3
**Presentation:** 3
**Contribution:** 2
**Rating:** 6
**Confidence:** 2

**Summary:**

This paper proposes an RL framework for geometric reasoning, named GCPO (Group Contrastive Policy Optimization). Specifically, it builds upon the verifiable reward in GRPO by introducing a group contrastive masking mechanism that assigns positive, negative, or zero rewards to “auxiliary drawing” actions depending on whether they contribute to the final correctness.

**Strengths:**

1. Clarity of writing: The paper is well written and easy to follow, with a clear logical flow.

2. Clarity of the proposed method: GCPO is conceptually straightforward and easy to implement while remaining effective.

3. Comprehensive experimental evaluation: The experiments are extensive and demonstrate clear effectiveness.

**Weaknesses:**

1. Limited OOD evaluation: The OOD datasets contain only a small number of samples, which makes it difficult to convincingly demonstrate the generalization ability of the proposed method beyond the training distribution.

2. Lack of critical ablation studies: Some key hyperparameters and design factors are not systematically analyzed, such as the KL coefficient, sampling temperature/decoding strategy, and the quantitative relationship between positive/negative masking ratios and performance.

**Questions:**

See weaknesses

---

### Official Review · Reviewer_aY5U · 2025-11-02

**Soundness:** 3
**Presentation:** 4
**Contribution:** 2
**Rating:** 6
**Confidence:** 4

**Summary:**

The authors introduce Group Contrastive Policy Optimization (GCPO) as a novel
reinforcement learning mechanism for LLMs to improve geometric reasoning. The
method includes a auxiliary reward, group contrastive masking and a length
reward. Experimental results show that GCPO outperforms GRPO and ToRL models.

**Strengths:**

S1: GCPO outperforms the existing GRPO and ToRL reinforcement learning frame-
works.

S2: GCPO can be applied to relatively small models (e.g., 1.5GB, 3GB and 7GB).

S3: The ablation study and discussion in the paper is interesting.

**Weaknesses:**

W1: The proposed modifications to GRPO are inspired by other, existing
techniques and thus the novelty is somewhat limited. For example, the length
reward was introduced earlier.

**Questions:**

N/A

---

### Note · Authors · 2025-12-24

I have read and agree with the venue's withdrawal policy on behalf of myself and my co-authors.